# Heart Failure with Reduced Ejection Fraction: The Role of Cardiovascular and Lung Ultrasound beyond Ejection Fraction

**DOI:** 10.3390/diagnostics13152553

**Published:** 2023-07-31

**Authors:** Syuzanna Shahnazaryan, Sergey Pepoyan, Hamayak Sisakian

**Affiliations:** Clinic of General and Invasive Cardiology, “Heratsi” Hospital Complex #1, Yerevan State Medical University, 2 Koryun Street, Yerevan 375025, Armenia; syuzanna.shahnazaryan@gmail.com (S.S.); spepoyan@gmail.com (S.P.)

**Keywords:** heart failure with reduced ejection fraction, echocardiography, outpatient monitoring, left ventricular filling pressure, lung ultrasound, inferior vena cava

## Abstract

Heart failure with reduced ejection fraction (HFrEF) is considered a major health care problem with frequent decompensations, high hospitalization and mortality rates. In severe heart failure (HF), the symptoms are refractory to medical treatment and require advanced therapeutic strategies. Early recognition of HF sub- and decompensation is the cornerstone of the timely treatment intensification and, therefore, improvement in the prognosis. Echocardiography is the gold standard for the assessment of systolic and diastolic functions. It allows one to obtain accurate and non-invasive measurements of the ventricular function in HF. In severely compromised HF patients, advanced cardiovascular ultrasound modalities may provide a better assessment of intracardiac hemodynamic changes and subclinical congestion. Particularly, cardiovascular and lung ultrasound allow us to make a more accurate diagnosis of subclinical congestion in HFrEF. The aim of this review was to summarize the advantages and limitations of the currently available ultrasound modalities in the ambulatory monitoring of patients with HFrEF.

## 1. Introduction

Heart failure (HF) is considered a major health care problem with more than a million new cases every year [1]. Over the last few decades, guideline-recommended treatment has considerably improved the outcomes of heart failure with reduced ejection fraction (HFrEF) [2,3]. Despite this, the mortality and rehospitalization rates remain high [1,2,4,5]. Moreover, the number of hospitalizations for HF is expected to increase significantly in the future and may double by 2045 due to the aging and growth of the population [6,7].

HF decompensation may be provoked by trigger factors such as infection, myocardial ischemia, acute renal injury, or anemia. In the case of disease deterioration due to increasing fluid retention, subclinical congestion precedes the clinical manifestation of HF symptoms and hospitalizations for an acute HF decompensation by several days or weeks. Therefore, the identification of a vulnerable period before the symptomatic HF decompensation allows for a prompt increase in diuretic doses and treatment modification, which may prevent the upcoming hospital admission [8]. Physical examination, particularly the identification of crackles on auscultation, is the main strategy routinely used for the prediction and diagnosis of left-sided HF decompensation in outpatients [9]. However, while it allows for the identification of symptomatic patients, the asymptomatic phase of HF decompensation is usually missed.

Hospitalization negatively affects the prognosis and quality of life [10]. Moreover, it is strongly associated with a risk of cardiovascular and all-cause mortality, which increases progressively with every subsequent hospitalization [11]. Thus, the prevention of rehospitalization is one of the main goals of HF management. For an effective reduction in unexpected hospital visits, early prediction of HF decompensation and prompt treatment modification are of paramount importance. Effective ambulatory monitoring and timely treatment modification are the key strategies for the reduction in cardiovascular events and improvement in the prognosis [12]. Numerous data suggest that HF patients who have undergone an echocardiographic examination have better survival rates due to more intensive medical treatment and interventions [13].

The aim of this review was to summarize the current knowledge regarding echocardiographic and ultrasound features of subclinical congestion among a broad range of HFrEF patients.

## 2. Left Ventricular Decompensation: Natriuretic Peptides and Basic Echocardiography

It is evident that hemodynamic deterioration precedes the development of clinical signs and symptoms by days or weeks, thus symptomatic clinical congestion may be seen as the “tip of an iceberg” of hemodynamic compromise [14]. Due to the delay between symptom manifestation and the progressive left ventricular (LV) filling pressure increase, physical examination alone may be insufficient for the identification of patients in the vulnerable phase of asymptomatic decompensation and for the prediction of upcoming hospital readmissions in outpatients [15,16].

It is well-established that an increased venous pressure can be detected by cardiovascular ultrasound, particularly by the assessment of inferior vena cava (IVC) characteristics and the detection of interstitial pulmonary edema through B-line identification. On the other hand, intracardiac hemodynamic changes and LV filling pressure increase precede the manifestation of HF symptoms and can be diagnosed by echocardiography [12,17]. Comprehensive cardiovascular ultrasound with tissue Doppler imaging (TDI) and lung ultrasound overcome the limitations of physical examination and clinical picture-based strategies for the prediction of HF decompensation.

Over the last decade, it has become apparent that natriuretic peptide-guided management of ambulatory chronic HF patients may optimize the treatment. Natriuretic peptides are important components of HF diagnostic work-up due to their high negative predictive value. However, the data on the utility of natriuretic peptides in the prevention of HF decompensations are relatively scarce. In the PARADIGM-HF trial, low levels of natriuretic peptides were associated with lower risks of cardiovascular events [18]. In the GUIDE-IT study involving 894 high risk patients with HFrEF, the most important predictor of cardiovascular death and HF hospitalizations was the baseline natriuretic peptide level [19]. The data suggest that natriuretic peptides also correlate with the risk of hospitalization and the duration of hospital stay [6]. However, with the introduction of a neprilysin inhibitor in the treatment of HFrEF, the accuracy and reliability of natriuretic peptides as prognostic markers have been questioned. By inhibiting neprilysin, sacubitril leads to an increase in B-type natriuretic peptide (BNP) and a reduction in the N-terminal pro-B-type natriuretic peptide (NT-proBNP) concentrations [20]. Thus, the role of natriuretic peptides in the prediction of hospital admissions in outpatients has to be further investigated.

Another concern regarding the diagnostic role of natriuretic peptides is related to morbidly obese patients, who have consistently lower natriuretic peptide levels [21,22,23]. Other limitations of natriuretic peptides include their level changes in atrial fibrillation and renal dysfunction. Given the limitations of natriuretic peptides, along with the complex physiology of HF, it is reasonable to consider the implementation of other parameters and markers for the improvement of HF management and the prevention of decompensation.

Basic echocardiographic assessment is an important tool for the diagnosis, classification, and treatment of HF. Some authors recommend using LV ejection fraction (EF) for the assessment of prognosis in ambulatory HFrEF patients, based on the role of EF dynamic increase as a predictor marker of an improved survival and hospitalization risk in this patient population [24]. However, EF loses its prognostic role at later stages of the disease and when the values are close to the threshold [25]. As a prognostic marker for HF decompensations, EF has other important limitations. Among them, technical pitfalls, preload dependence, possible measurement errors in the presence of severe mitral regurgitation, atrial fibrillation or heart rate variations, and a weak association with functional class and exercise capacity [26,27,28]. The above-mentioned limitations advocate for the use of additional parameters and methods for the assessment of LV function such as TDI, longitudinal strain quantification by speckle tracking, 3-dimensional echocardiography, and cardiac magnetic resonance imaging (MRI).

An increase in left atrial (LA) volume and LV filling pressure contribute to LA remodeling and HF decompensation, leading to frequent hospitalization due to the lung congestion. In clinical practice, the LA volume index is widely used to assess LA overload. A clinically significant increase is considered an LA volume index >34 mL/m^2^ [29]. The LA volume index is an independent predictor of cardiovascular events including atrial fibrillation as well as cardiovascular mortality [26,30]. Moreover, the LA volume index is a marker of diastolic dysfunction and, unlike EF, correlates well with the patient’s functional class and overall exercise capacity [27,31]. The LA volume index correlates with the E/e’ ratio and can serve as a useful predictor of increased LV filling pressure during exercise [32].

Echocardiography with intracardiac hemodynamic assessment provides a semiquantitative analysis of decompensation and identifies filling pressure abnormalities. LV filling pressure measurements are of special interest in patients with HFrEF because the increase in filling pressure underlines the pathophysiological mechanisms of HF deterioration and precedes the manifestation of HF signs and symptoms [12].

Currently, echocardiographic Doppler and TDI are widely used in clinical practice for the evaluation of LV systolic and diastolic functions. As a marker of diastolic dysfunction, LV filling pressure can be assessed by calculating the E/e’ ratio, where the E wave is the peak velocity of the early diastolic flow across the mitral valve, as measured by pulsed-wave Doppler, and e’ is the early diastolic velocity of the septal or lateral mitral annulus obtained by TDI (Figure 1). The accepted threshold for the normal LV end-diastolic pressure is E/e’ <8, while E/e’ >14 and peak tricuspid regurgitation (TR) velocity >2.8 m/s indicate high LV end-diastolic pressure [33]. Recent studies emphasize the importance of the routine measurements of E/e’ for the assessment of LV filling pressure, risk stratification, and prognostication in the HF patient population [33,34]. Benfari G. et al. suggested an E/e’ ratio >14 as a cut-off for the identification of high-risk patients [35]. The authors reported increased short- and long-term mortality in individuals with E/e’ >14, with a considerable worsening of prognosis in E/e’ >20 [35]. Therefore, in HFrEF, additional parameters should be considered for the assessment of LV filling pressures and atrial hypertension to predict intracardiac volume overload and subclinical deterioration.

In summary, a mean E/e’ ratio >14 and LA volume index >34 mL/m^2^ are important markers for the assessment of LV diastolic dysfunction and correlate with invasively measured LV filling pressure [36,37,38]. Moreover, unlike EF, the LA volume index, E/e’, and TR velocity have better correlation with overall physical capacity in patients with HFrEF [33].

The systemic review and meta-analysis of the echocardiographic quantification of LV filling pressure found that the non-invasive echocardiographic estimation of filling pressure correlated with the invasive right heart catheterization results and its use was feasible in both HF with reduced and preserved EF [39]. Although in patients with HFrEF the pooled correlation coefficient was stronger for clinical translation, the application of E/e’ along with LA volume and TR velocity showed increased correlation with the invasive assessment [40,41].

HF patients with LVEF below 40% may demonstrate different echocardiographic patterns of congestion based on the LV filling pressure, LA volume index, and lung ultrasound. The implementation of these parameters in routine clinical practice may provide important information and be of value in predicting HF decompensation [42,43].

Ballo R. and co-authors investigated the prognostic role of longitudinal LV systolic function assessed by TDI. The authors evaluated the average value of lateral and septal peak systolic annular velocities (Sm) and reported the Sm threshold of ≤6.5 cm/s as a statistically significant prognostic value in dilated cardiomyopathy of non-ischemic origin [44]. However, the patients with an ischemic etiology of HF were not included in the study, thus, the possible influence of local asynergies on the value of the peak systolic velocity of the mitral ring was not investigated.

Despite several limitations of single E/e’ ratio measurements in the prediction of elevated LV filling pressure, the complex assessment of LV filling pressure, pulmonary artery systolic pressure, and LA volume index has a significant diagnostic value. When combined, these parameters allow for risk stratification and are associated with outcomes in patients with heterogeneous EF within the HFrEF population.

## 3. Speckle Tracking

Over the last decades, speckle tracking echocardiography (STE) has emerged as a relatively new technique for the assessment of HF individuals. STE provides information about the ventricular function by identifying and analyzing the movement of dots or “speckles” in the ventricular wall image, and by tracing the length and thickness changes in any given segment of the myocardium. Evaluation of the myocardial displacement is performed in 4-chamber, 2-chamber, and apical long-axis views and is presented as a percentage. Speckle tracking imaging allows for the measurement of the global longitudinal strain (GLS), radial, and circumferential strains of the ventricles as well as atrial longitudinal strain.

To date, the LV GLS is the most widely investigated and implemented parameter among the speckle tracking strains. Numerous data suggest that GLS may be a useful marker for the clinical risk stratification and prognostication of HFrEF [45,46,47]. As a prognostic marker, GLS is superior to LV EF in predicting reduced exercise capacity, poor prognosis, and mortality in patients with HFrEF [48,49,50]. Compared to LV EF, GLS allows for a more accurate quantification of the extent of systolic dysfunction. Other significant disadvantages of LV EF include the inability to quantify regional myocardial function and geometric assumptions [51].

Carluccio E. and co-authors analyzed the role of LA longitudinal strain in HFrEF patients. Based on the results, peak LA longitudinal strain has a significant prognostic value in HFrEF, independent of LA volume and LV longitudinal contraction [52]. Moreover, reduced peak LA longitudinal strain is associated with higher LA and LV volumes as well as more severe systolic and diastolic dysfunctions [52]. In HF patients, LA longitudinal strain can also be used as an alternative parameter for the assessment of LV filling pressure [53,54].

STE is superior to standard echocardiography in detecting impaired systolic function using longitudinal and circumferential deformation, particularly in the heterogeneous group of HF patients. Since its initial use over 15 years ago, STE has emerged as a method that provides more robust, reproducible, and sensitive parameters of regional dysfunction and myocardial dyssynchrony. An important advantage of STE imaging over TDI is the absence of the angle-dependency [27]. However, STE requires high quality images to obtain accurate measurements and is a time-consuming technique. Another limitation of STE is the need for high resolution images, which are not always possible to obtain during routine examination. Therefore, taking into consideration the number of limitations, STE is not always applicable for the assessment of ambulatory patients.

## 4. Lung Ultrasound

Lung ultrasound (LUS) is a novel strategy for the assessment of pulmonary congestion in HF patients. Over the last few years, it has been widely used for the differential diagnosis of acute onset dyspnea in emergency situations and for the management of acute HF. Additionally, LUS has a prognostic value in the monitoring of chronic HF patients.

In healthy lungs, only horizontal parallel A-lines can be observed. In the case of extravascular fluid accumulation in the lungs, B-lines can be detected, which are vertical white artifacts arising from the pleura and reaching the bottom of the screen (Figure 2) [55]. LUS can be performed in the supine position of the patient in 28 areas and the total number of B-lines is reported [55]. Progressive increase in B-lines is associated with the severity of pulmonary congestion, the total number of B-lines >5 in 28 zones is considered abnormal, while B-lines >30 suggest severe congestion [55,56].

LUS has a high prognostic value for the prediction of hospitalization. Among the patients hospitalized for acute HF, the presence of residual B-lines at discharge is associated with a higher risk of rehospitalization [57,58]. LUS can also be effectively used as a prognostic marker of hospital readmission in ambulatory patients with chronic HF. There is a strong association between the presence of B-lines (greater than 15 in 28 zones) and an increased risk of hospitalization for HF decompensation [59,60]. Moreover, LUS is superior and more sensitive in the detection of pulmonary congestion compared to physical examination. An important advantage of LUS over physical examination is the ability to detect asymptomatic patients with subclinical congestion both at discharge and in outpatient settings [55].

The use of LUS for the detection of residual congestion at discharge among the patients admitted for acute decompensated HF allows for the identification of undertreated patients who require more intensive diuretic regimens for the prevention of early rehospitalizations. Additionally, the detection of asymptomatic pulmonary congestion before the symptom onset allows for an early diagnosis of subclinical HF decompensation in outpatients. Thus, it may enable the making of timely treatment modifications, particularly loop diuretic dose up-titration, for the prevention of HF decompensation in ambulatory settings.

In summary, the detection of B-lines by LUS plays an important role in the differential diagnosis of congestion and acute decompensated HF. Moreover, implementation of LUS in the ambulatory follow-up may enable HF specialists to identify the patients at high risk of an upcoming decompensation and may provide guidance in determining the optimal treatment strategy for the prevention of hospitalization among patients with chronic HF.

## 5. Right Ventricular HF Decompensation

Both patients and health care providers can be involved in the clinical follow-up-based approach for the prevention of hospitalization in ambulatory patients. Self-assessment of congestion by the patient requires daily weight checking and self-administration of an additional diuretic dose in the case of body weight increase for ≥2 kg in 2 days or the exacerbation of edema [61,62]. Despite the improvement in the outcomes, the self-control strategy highly depends on the compliance and psychological state of the individual [63]. Moreover, ambulatory weight change monitoring and self-assessment of the symptoms are unreliable and have low effectiveness in the prevention of HF hospitalizations [64]. This is due to the fact that the development of edema and HF decompensation are driven not only by the circulating blood volume increase, but also largely depend on fluid redistribution in the body [65].

In patients with HF, echocardiographic measurements of right ventricular (RV) pump function and right atrial (RA) pressure provide important information about RV dysfunction, overload, pulmonary artery, and central venous pressures. For the evaluation of RV systolic function, EF and tricuspid annular plane systolic excursion (TAPSE) are used in clinical practice. However, both parameters have important limitations. Particularly, the assessment of RV EF is difficult due to RV geometry and suboptimal visualization of the endocardial borders. Limitations of TAPSE include angle and load dependence, and extrapolation of single segment displacement amplitude to the overall RV ventricular function [66]. Therefore, the implementation of additional parameters and advanced techniques for the evaluation of RV systolic function should be considered in the monitoring of HFrEF patients.

New parameters for the quantitative measurement of RV function have been widely investigated. Based on the available data, they are reliable, have a good correlation with the cardiac MRI findings, and can be used for the prediction of RV remodeling and clinical deterioration of the patients [66,67]. TAPSE, RV shortening, and peak systolic tricuspid annular velocity (S’) measured by TDI have an important role in the assessment of HF severity and the identification of patients with a high risk of frequent rehospitalization and mortality [68].

The prognostic value of RA enlargement was analyzed in the study conducted by Almodares Q. et al. [69]. Meanwhile, RV dysfunction and RA pressure increase were reported as poor outcome predictors and were associated with higher event risk among ambulatory patients with HF at 12 months after discharge [70].

Studies have demonstrated that patients with an IVC diameter >2.5 cm and collapse of <50% on inspiration have an RA pressure of more than 10 mmHg (Figure 3) [71,72]. The evaluation of IVC can be easily performed and successfully used in outpatients with HF [73]. The examination can be reliably performed by medical personal with limited training and experience in echocardiography, particularly by point-of-care mobile ultrasound systems [74,75,76]. The assessment of IVC diameter and collapsibility in ambulatory patients may aid in the prediction of HF deterioration and hospitalization, and may have an added prognostic value when combined with physical examination [77,78]. In the HF population, IVC distension correlated with an increased risk of adverse events; furthermore, as a poor outcome predictor, its value was comparable with NT-proBNP [45]. Among the individuals with HF, the presence of dilated IVC was associated with a higher risk of complications, particularly renal function decline [79]. However, IVC dilation develops after significant expansion of the plasma volume and rise in the RA pressure, thus, it cannot serve as an early sign of congestion. Moreover, the limitations of IVC assessment include the difficulty of measurement interpretation in the case of intrathoracic pressure increase, intraabdominal hypertension, and positive pressure mechanical ventilation.

A summary of the previous studies on the role of cardiovascular and lung ultrasound parameters in the assessment of congestion and prognosis of HFrEF patients is presented in Table 1.

In this review, we outlined the gaps in the LV EF evaluation by basic echocardiography and emphasized the importance of implementing other parameters beyond EF for the evaluation of HF patients in everyday clinical practice. This paper focused on the heterogeneous group of HFrEF patients with high or normal LV filling pressure and RV congestion. We suggest performing a risk stratification of HFrEF patients based on the echocardiographic evidence of LA hypertension for the prediction of decompensation and treatment optimization.

## 6. Future Directions

Cardiovascular and lung ultrasound with hemodynamic assessment is a low cost comprehensive tool, which should be more widely used in heart failure clinics for the early detection of congestion signs before the development of clinical deterioration. The evidence suggests that subclinical congestion may be present in most of the clinical scenarios in both the outpatient period and after hospital discharge. The review of the currently accepted prognostic markers shows a limited significance of each of these parameters in clinical and predictive assessment.

The assessment of congestion by several radiation-free, not time-consuming ultrasound methods may guide the intensification of diuretic treatment and decisions regarding the frequency of follow-up visits.

## 7. Conclusions

Several important echocardiographic parameters beyond EF can be routinely used as part of an algorithm for the prediction of decompensation in chronic HF patients.Integration of multiple parameters, which can be easily determined by lung ultrasound and cardiovascular ultrasound including LV filling pressure, central venous pressure, left and right atrial remodeling enables moving beyond considering only LV systolic function.Complex cardiovascular ultrasound assessment allows one to obtain more precise parameters of HF deterioration and enables the identification of patients who require a personalized treatment approach.

## Figures and Tables

**Figure 1 diagnostics-13-02553-f001:**
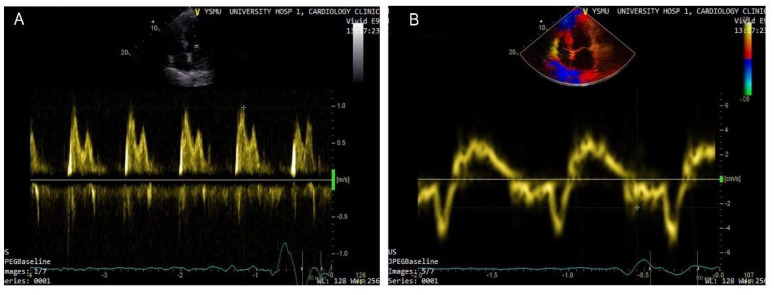
Doppler-based assessment of LV filling pressure. Raised filling pressure based on increased E/e’ ratio derived from the spectral Doppler of transmitral flow early peak velocity (E wave) (arrow) (**A**) and tissue Doppler of the mitral annular displacement early diastolic peak velocity (e’ wave) (arrow head) (**B**); E/e’ > 15.

**Figure 2 diagnostics-13-02553-f002:**
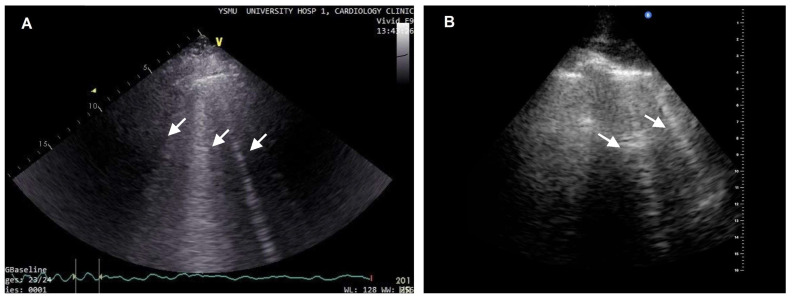
(**A**) Two to three moderate B-lines in the lung ultrasound (arrows); (**B**) Wide B-lines (arrows) detected by bedside point-of-care ultrasound in a patient with acute decompensated HF in CCU.

**Figure 3 diagnostics-13-02553-f003:**
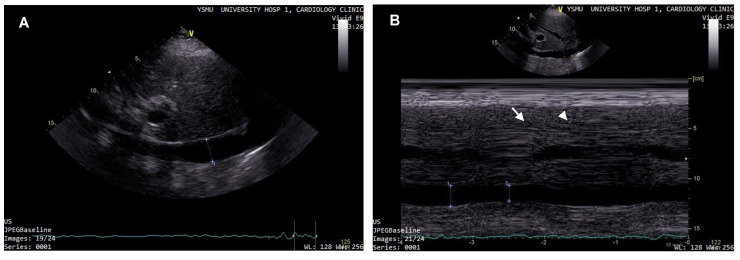
(**A**) Increased IVC diameter in a patient with high right atrial pressure; (**B**) IVC diameter on expiration (arrow), and an impaired collapse (<50%) on inspiration (arrow head).

**Table 1 diagnostics-13-02553-t001:** Review of previous studies on the role of cardiovascular and lung ultrasound parameters in the assessment of congestion and the prognosis of patients with acute and chronic HFrEF.

Method of Evaluation	Study Type	Patient Characteristics	Limitations and Pitfalls	Results	Authors
EF	Prospective, multivariate, propensity match	799 HF patients with decompensation.	Hospital based, selection bias to perform echocardiography (19% without echo).	Better quality of care and prognosis.	Tribouilloy C., Rusinaru D., Mahjoub H. et al. (2008) [13]
BNP	Prospective	316 patients with systolic HF: obese vs. lean.	Selected population with advanced HF.	BNP levels are lower in obese and overweight patients compared to lean patients.	Horwich T., Hamilton M., Fonarow G. et al. (2006) [22]
GLS	Large scale retrospective	1102 HFrEF patients.	NYHA were not assessed; single cardiac cycle, GLS assessment.	GLS is a strong prognosticator of all-cause mortality; GLS was superior to EF.	Sengeløv M., Jørgensen P.G., Jensen J. et al. (2015) [49]
LA and RA	Retrospective	289 elderly patients with HF	Retrospective study, elderly population	RA and LA remodeling independently of LV EF are associated with all-cause mortality.	Almodares Q., Guron C.W., Thurin A. et al. (2017) [69]
LUS	Multicenter, randomized	130 patients with acute HF.	No assessment of long-term hospitalization.	Decrease in hospitalization in the LUS-guided group.	Marini C., Fragasso G., Italia L. et al. (2020) [42]

EF—ejection fraction; HF—heart failure; BNP—B-type natriuretic peptide; GLS—global longitudinal strain; HFrEF—heart failure with reduced ejection fraction; NYHA—New York Heart Association; LA—left atrium; RA—right atrium; LV—left ventricle; LUS—lung ultrasound.

## Data Availability

Data sharing not applicable.

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
