# Peer review of "Heart Failure with Reduced Ejection Fraction: The Role of Cardiovascular and Lung Ultrasound beyond Ejection Fraction"

_diagnostics, 2023, doi:10.3390/diagnostics13152553_

Round 1

Reviewer 1 Report

This is a nice writing manuscript. The methodology and the structure of the manuscript are correct. The quality of the images is good and help the readers to understand better the topic.

This review aims to consolidate the present understanding of the echocardiographic and ultrasound features related to subclinical congestion in a diverse spectrum of patients diagnosed with HFrEF.

The last review in the area of utility of lung ultrasound in cardiac disease was at least 5 years ago so it is relevant in the field. This manuscript is a useful update in this area.

In general, is a well-written manuscript with correct methodology. The authors must consider aiding a section ‘future directions.’

The conclusions are consistent with the evidence and arguments presented and they address the main question posed.

The are references appropriate

The authors must consider aiding tablets with the clinical characteristics of the mentioned studies in order to make the discussed subject more understandable.

Good quality of the existing figures.

Author Response

Please see the attachment․

Reviewer 2 Report

The Authors present the use of advanced  cardiovascular ultrasound modalities in  patients with HFrEF. The Paper is interesting  and well written . The Authors show the  role of basic echocardiography  in relation with  BNP, the hemodynamic assessment  with E/e' and left atrial dimension and function   in relation with  left ventricular filling pressure, speckle tracking  with GLS  (superior to EF in prognosis) ,  lung ultrasound use in patients with pulmonary congestion  . The paper summarize in elegant way clinical  application of these new techonologies.

I 'm not in agreement with  line 203-204 about high interobserer variability of speckle tracking. Different Authors presented lower interobsever variability with Speckle trackling.( Di Salvo et al Eur J of Echocardiogr 2008;9:754-6)

Mistakes

Line 123 doppler to change in Doppler 

The Legenda of Photos need a better description

Line 224-225  legenda Fig 2 to put arrows on B line

Line 302-303  legenda Fig 3 to  put  arrows evident in inspiration and espiration 

Line 140-141  legenda Fig 1: to put arrows  on  E and e' ,  to report in legenza the value  of E/e' ratio 

The references do not follow regulation of Journal

There is a references file

The English language  is correct
